# Phase I Trial of Prophylactic Donor-Derived IL-2-Activated NK Cell Infusion after Allogeneic Hematopoietic Stem Cell Transplantation from a Matched Sibling Donor

**DOI:** 10.3390/cancers13112673

**Published:** 2021-05-28

**Authors:** Raynier Devillier, Boris Calmels, Sophie Guia, Mohammed Taha, Cyril Fauriat, Bechara Mfarrej, Geoffroy Venton, Eric Vivier, Daniel Olive, Christian Chabannon, Didier Blaise, Sophie Ugolini

**Affiliations:** 1Hematology Department, Institut Paoli-Calmettes, 13009 Marseille, France; blaised@ipc.unicancer.fr; 2Immunity and Cancer Team, CRCM, Inserm Institut Paoli-Calmettes, Aix-Marseille Université, 13007 Marseille, France; mohammed.taha@inserm.fr (M.T.); cyril.fauriat@inserm.fr (C.F.); daniel.olive@inserm.fr (D.O.); 3CRCM, Inserm Institut Paoli-Calmettes, Aix-Marseille Université, 13009 Marseille, France; calmelsb@ipc.unicancer.fr (B.C.); mfarrejb@ipc.unicancer.fr (B.M.); CHABANNONC@ipc.unicancer.fr (C.C.); 4Module Biothérapies du Centre d’Investigations Cliniques de Marseille, Inserm Institut Paoli-Calmettes, Aix-Marseille Université, 13007 Marseille, France; 5Centre d’Immunologie de Marseille-Luminy CIML, CNRS, INSERM, Aix Marseille Université, 13009 Marseille, France; guia@ciml.univ-mrs.fr (S.G.); geoffroy.venton@ap-hm.fr (G.V.); vivier@ciml.univ-mrs.fr (E.V.); 6Assistance Publique des Hôpitaux de Marseille, Hôpital de la Timone, 13005 Marseille, France; 7Innate Pharma, 13276 Marseille, France

**Keywords:** allogeneic hematopoietic stem cell transplantation, cellular immunotherapy, IL-2-activated NK cells, antitumor immunity

## Abstract

**Simple Summary:**

Allogeneic hematopoietic stem cell transplantation (allo-HSCT) is a curative option for high-risk hematologic malignancies. However, disease recurrence after allo-HSCT remains a critical issue, underlining the need to develop maintenance therapy. In this context, NK cell-based immunotherapies could enhance graft-versus-tumor effect without triggering graft-versus-host disease. In this prospective phase I clinical trial, we demonstrated the safety of donor-derived NK cell infusion as a prophylactic treatment after allo-HSCT for patients with hematological malignancies. This opens perspectives for future developments of NK cell based therapeutic strategies after allo-HSCT with low incidence of GVHD, representing an advantage over post-transplant T cell modulations that are commonly used in clinical routine.

**Abstract:**

*Background*: NK cell-based immunotherapy to prevent relapse after allogeneic transplantation is an appealing strategy because NK cells can provide strong antitumor effect without inducing graft-versus-host disease (GVHD). Thus, we designed a phase-I clinical trial evaluating the safety of a prophylactic donor-derived ex vivo IL-2 activated NK cell (IL-2 NK) infusion after allo-HSCT for patients with hematologic malignancies. *Methods*: Donor NK cells were purified and cultured ex vivo with IL-2 before infusion, at three dose levels. To identify the maximum tolerated dose was the main objective. In addition, we performed phenotypical and functional characterization of the NK cell therapy product, and longitudinal immune monitoring of NK cell phenotype in patients. *Results*: Compared to unstimulated NK cells, IL-2 NK cells expressed higher levels of activating receptors and exhibited increased degranulation and cytokine production in vitro. We treated 16 patients without observing any dose-limiting toxicity. At the last follow up, 11 out of 16 treated patients were alive in complete remission of hematologic malignancies without GVHD features and immunosuppressive treatment. *Conclusions*: Prophylactic donor-derived IL-2 NK cells after allo-HSCT is safe with low incidence of GVHD. Promising survivals and IL-2 NK cell activated phenotype may support a potential clinical efficacy of this strategy.

## 1. Introduction

Allogeneic hematopoietic stem cell transplantation (allo-HSCT) is a curative option for high-risk hematologic malignancies. Many improvements have been made in reducing transplant-related toxicity and mortality over the past 20 years while disease recurrence after allo-HSCT remains a critical issue. Numerous strategies are being developed to prevent relapse after allo-HSCT such as maintenance treatments with cellular immunotherapies, tyrosine kinase inhibitors, and/or immunomodulatory drugs [1,2,3,4].

Donor lymphocyte infusion (DLI) is commonly used as a cellular immunotherapy after allo-HSCT to enhance T cell-mediated graft-versus-tumor (GVT) effect. Initially, DLI utilization was reported to treat hematologic relapse after allo-HSCT with limited efficacy [5,6,7]. More recently, the use of DLI to treat molecular relapse and/or mixed chimerism (preemptive therapy) demonstrated improved results, suggesting that cellular immunotherapy after allo-HSCT is more effective when used to clear residual disease [7]. These studies paved the way for the use of prophylactic DLI. Both ourselves and others have shown that while prophylactic DLI may produce promising outcomes it is also associated with a high rate of graft-versus-host disease (GVHD) [7,8,9,10], due to alloreactive T cells. However, allogeneic Natural Killer (NK) cell infusions may exhibit GVT effect without inducing GVHD [11,12,13,14]. This is supported by both clinical and biological evidences from the recent development of allogeneic NK cell-based therapy [15,16,17].

NK cells are cytotoxic innate lymphoid cells involved in tumor immunosurveillance, whose cell-mediated killing of target cells is controlled by a series of activating and inhibitory receptors. Inhibitory receptors include MHC class I-specific killer Ig-like receptors (KIRs) and CD94/NKG2A [18,19]. Target cells can directly stimulate NK cell activation through the engagement of NK cell-activating receptors such as natural cytotoxicity receptors (NKp46/NCR1, NKp44/NCR2, NKp30/NCR3), NKG2D, and DNAM-1. NK cells are also able to detect antibody-coated cells via the CD16 (FcγRIIIA) cell surface receptor. Upon activation, NK cells can kill target cells and produce cytokines, such as IFN-γ that modulate other immune cell functions [18].

It was previously shown that early recovery of NK cell counts after allo-HSCT is associated with a lower incidence of both relapse and GVHD [20,21]. However, newly generated NK cells still have low activation and killing capabilities and harbor maturation defects during the first months after allo-HSCT [22,23,24,25,26]. Enhancing NK cell activity in patients early after allo-HSCT by infusing activated donor-derived NK cells could prevent disease relapse, which usually occurs in a median time of 4 to 6 months post-transplant. Here, we provide proof-of-concept, via mouse model, of the antitumor role of IL-2-activated NK cell infusion and we describe a phase I clinical trial that explores the feasibility and safety of donor-derived IL-2-activated NK (IL2-NK) cell infusion from an HLA-matched sibling donor (MSD), 60 to 120 days after allo-HSCT.

## 2. Materials and Methods

### 2.1. Study Design

The primary objective of this prospective monocentric phase 1 trial (NCT01853358) was to assess the safety of prophylactic donor-derived ex vivo purified and activated NK cell infusion 60 to 120 days after allo-HSCT from an MSD (Figure 1). The study was approved by the Paoli-Calmettes institutional review board, the Comité de Protection des Personnes (CPP), and the French competent authority (ANSM) in compliance with national regulations. Both recipients and donors were enrolled after signing a written informed consent form.

An initial phase of dose escalation (3+3 method) was planned to determine the maximum tolerated dose (MTD) using 3 dose levels: 1.0 (dose 1), 5.0 (dose 2), and 5.0–50 (dose 3) million cells/kg body weight (Figure 1). Dose-limiting toxicity (DLT) was defined as the emergence of grade 3–4 toxicity (including grade 3-4 severe acute and chronic GVHD) considered to be related to NK cell infusion within 30 days post infusion. A second phase allowed for the treatment of additional patients, with a final total of 10 patients treated at the MTD.

### 2.2. Selection Criteria

Patients were eligible according to the following inclusion criteria: age 18 to 70 years; allo-HSCT from an MSD for hematologic malignancy; and a conditioning regimen based on combining fludarabine, busulfan, and antithymocyte globulin (ATG). Exclusion criteria were a history of grade ≥ 2 acute GVHD (modified Glucksberg classification) and/or the use of systemic steroids at a dose ≥ 0.5 mg/kg prednisone or prednisolone at the time of NK cell infusion. Eligible patients were scheduled to receive a single donor-derived IL-2 NK cell infusion between day+60 and day+120 after allo-HSCT. In addition, a historical control cohort of patients treated in our center was retrospectively analyzed to evaluate the potential effect of IL2-NK cell infusion on clinical outcome. Selection criteria for this historical control group were: first allo-HSCT from an MSD for hematological malignancies, conditioning regimen based on fludarabine, busulfan, and ATG, and intermediate disease risk index (DRI), and no NK cell infusion or conventional DLI.

### 2.3. NK Cell Manufacturing and Treatment Modalities

Unstimulated mononuclear cells were collected from the original MSD as soon as possible near day+60 after allo-HSCT. Using the COBE^®^ or OPTIA^®^ Spectra apheresis system (Terumo BCT, Lakewood, CO, USA), 3 blood volumes were systematically processed in order to collect a sufficient number of peripheral blood NK cells for further processing. Immunomagnetic separation with the Miltenyi Biotec CliniMACS^®^ Cell Selection System was used for CD3^+^ cell depletion and subsequent CD56^+^ cell enrichment, following the manufacturer’s recommendations (Miltenyi Biotech, Bergisch Gladbach, Germany). Selected NK cells were cultured at 37 °C with 5% CO_2_ for 7 days in gas-permeable cell culture bags (VueLife Teflon, American Fluoroseal, Gaithersbrug, MD, USA) in RPMI 1640 (Life Technologies, Carlsbad, CA, USA) supplemented with 10% good manufacturing practices (GMP) grade fetal bovine serum (HyClone^®^, Thermo Scientific, Logan, UT, USA) in the presence of 1,000 U/mL of IL-2 (Proleukin, Chiron Corp.) at the initial concentration of 0.5 × 10^6^ cells/mL. At day 7, cells were washed and resuspended in 5% human serum albumin. Cell products were assessed at each stage for CD56^+^ and CD3^+^ cells using a single-platform flow cytometry protocol. Release criteria included: total nucleated cell viability >70%, negative microbiological test result, CD56^+^ cell count ≥1 × 10^6^/kg for dose level 1 and ≥5 × 10^6^/kg for dose levels 2 and 3, and CD3^+^ cell count <5 × 10^4^/kg. Standardized quality controls were employed at all steps of the manufacturing process. After 7 days of culture, the NK cell content of the final product was adjusted to fit the planned dose level and infused fresh between day+60 and day+120 after allo-HSCT. If the total amount of NK cells after culturing did not reach the planned dose, the whole product was infused.

### 2.4. Phenotypic Analysis of Donor NK Cells before and after IL-2 Activation

Human NK cells were stained with the following antibodies: anti-CD56 (B159), anti-CD3 (SK7), anti-CD16 (3G8), anti-CD226/DNAM-1 (DX11), anti-CD253/TRAIL (RIK-2), anti-CD69 (FN50), anti-CD336/NKp44 (p44-8.1), and anti-CD314/NKG2D (1D11) from BD Biosciences; anti-CD158a/KIR2DL1 (143211) and anti-CD159c/NKG2C (134591) from R&D systems; anti-CD335/NKp46 (9E2) and anti-CD158e1/KIR3DL1 (DX9) from BioLegend; anti-CD337/NKp30 (AF29-4D12) and anti-CD94 (REA113) from Miltenyi Biotec; and anti-CD158b1,b2,j;KIR2DL2/L3/S2 (GL183), anti-CD159a/NKG2A (Z199), and anti-CD158a,h;KIR2DL1/S1 (EB6B) from Beckman Coulter. Cells were stained with LIVE/DEAD Fixable Dead Cell Stain Kit (Molecular Probes), sample acquisition was performed on an LSRII UV cytometer (BD Biosciences), data were analyzed with FlowJo software v10 (Tree Star), and NK cells were defined as live CD3^−^CD56^+^ NK cells.

### 2.5. Functional Analysis of Donor NK Cells before and after IL-2 Activation

As a control, NK cells (2 × 10^5^) were seeded in 96-well round bottom plates. When indicated, we added 5 ng/mL human IL-12 (R&D Systems) and 20 ng/mL human IL-18 (MBL), or 500 ng/mL ionomycin (Sigma-Aldrich) and 2 ng/mL phorbol 12-myristate 13-acetate (PMA, Sigma-Aldrich), or K562 tumor cells at a 1:1 effector to target (E:T) ratio. For anti-CD16 mAb, the flat bottom plastic plates (BD Biosciences) were coated with F(ab’)2 anti-mouse IgG (Beckman Coulter) and anti-CD16 (10 µg/mL, 3G8, Beckman Coulter). Cells were stimulated for 4 h in the presence of brefeldin A (Golgistop, BD Biosciences) plus anti-CD107a-FITC and CD107b-FITC monoclonal antibodies, surface stained, and processed for intracellular IFN-γ detection. NK cells are defined as live CD3^−^CD56^+^ NK cells.

### 2.6. Cell Lines

All tumor cell lines were provided by Innate Pharma (France). They are regularly assessed (MycoAlert, LT07-318, Lonza) and were mycoplasma-free. The human tumor cell lines (the chronic myelogenous leukemia K562 and the B-lymphoblastoid cell line 221 transfected with HLA-Cw3, 221/HLA-Cw3) were maintained in RPMI 10% FCS. The mouse T lymphoma cell lines (YAC-1 and RMA-S) were maintained in RPMI 10% FCS, supplemented with 50 μM of 2-mercaptoethanol (31350010, Gibco).

### 2.7. Mice

C57BL/6 (CD45.2^+^) and C57BL/6-CD45.1^+^ (B6.SJL-*Ptprc^a^Pepc^b^*/BoyCrl) mice were purchased from Charles River Laboratories and maintained for at least 1 week in the Centre d’Immunologie de Marseille Luminy’s mouse house facility under specific pathogen-free conditions with a standard 12 h/12 h light–dark cycle with food and water ad libitum. Female mice were used at 8 to 12 weeks of age. NKp46-DTR (NKp46*^iCre/iCre^*-Rosa26*^DTR^*) and TgKIR mice were generated at the CIML laboratory and previously described [27,28]. TgKIR-RAG^KO^ mice were obtained by crossing TgKIR mice with immunodeficient RAG-1^KO^ mice which lack T and B lymphocytes. All experiments were conducted in accordance with institutional committees (Comité d’éthique de Marseille 14) and French and European guidelines for animal care.

### 2.8. Production and Infusion of Mouse IL-2-Activated NK Cells

Splenocytes from C57BL/6 (CD45.1^+^) mice were used to isolate donor NK cells. NK cells were enriched by using the mouse NK Cell Isolation Kit II (Miltenyi Biotec) and activated in vitro for 5 days with 5.000 U/mL IL-2 (Chiron) before intravenous (IV) infusion into C57BL/6 (CD45.2^+^) recipient mice.

### 2.9. Mouse NK Cell Analyses Ex Vivo

Mice were perfused with PBS before sampling. The NK cell compartment (origin: CD45.1^−^ and CD45.2^+^) was then analyzed at different time points post infusion in the spleen, liver, lung, blood, and lymph nodes from recipient mice. In particular, the percentages of CD69^+^-activated CD45.1^+^ and CD45.2^+^ NK cells were monitored over time by flow cytometry. The NK cells were also incubated for 4 h at 37°C in the presence of monensin (1/1,000; BD), brefeldin A (1/1,500; BD), CD107a monoclonal antibodies, and co-cultured with either YAC-1 tumor cells at a 1:1 E: T ratio or with 25 ng/mL mouse IL-12 (eBiosciences) and 20 ng/mL mouse IL-18 (MBL). The IFN-γ production was assessed by intracellular staining. The LIVE/DEAD Fixable Dead Cell Stain Kit (Molecular Probes) and the following antibodies were used for staining: anti-CD3 (500A2), anti-CD19 (1D3), anti-NKp46 (29A1.4), anti-CD69 (H1.2F3), anti-NK1.1 (PK136), anti-CD107a (1D4B), anti-CD45.1 (A20), anti-CD45.2 (104), and anti-IFN-γ (XMG1.2), with Fc blocking (2.4G2) prior to staining.

### 2.10. Mouse Tumor Models

For RMA-S tumor experiments, 2.5 × 10^3^ cells were injected intravenously before infusion of mouse donor IL-2 NK cells (2–4 × 10^6^ cells). The day prior to tumor injection, NK cells were depleted by a single IV injection of 200 ng DT (Calbiochem) and maintained by additional injections. For 221/HLA-Cw3 tumor experiments, 6 × 10^6^ cells were injected intravenously before infusion of mouse donor IL-2 NK cells (5–10 × 10^6^ cells).

### 2.11. Immune Monitoring Post Donor-NK Cell Infusion in Patients

Blood samples were collected from patients at day 0 prior to IL-2 NK cell infusion and at days 1, 9, 16, 30, 90, and 180 after NK cell infusion (Figure 1). The PBMCs were isolated from these samples and frozen for subsequent FACS analyses including NK cell phenotype and antitumor functions. Functional assays were performed by co-culturing patient PBMCs with K562 in a 10:1 E:T ratio for 4 h in the presence of CD107a-staining antibodies plus brefeldin A (Golgistop, BD Biosciences). Intracellular staining was performed for IFN-γ production.

### 2.12. Statistical Analyses

The figure legends detail the statistical analyses for each relevant experiment. Normality and homogeneity of variance were assessed. Statistical tests were performed using Prism 7 (GraphPad Software) and R software 4.0.3 version (https://cran.r-project.org/; Accessed 28 May 2021). Kaplan–Meier calculations were used to analyze mouse survival curves. Statistical significance was set at *p* value <0.05 (*p* ≥ 0.05 not significant, ns); values are included in the figures.

A semi-landmark model was used to compare clinical outcomes of the patients included in the DLI-NK trial (IL-2 NK group) to the historical cohort (control group). The cumulative incidence of GVHD and survivals were computed from the time of IL-2 NK cells for patients in the clinical trial. For the control group, GVHD and survivals were computed from day+90 after allo-HSCT (median time between allo-HSCT and IL-2 NK cell infusion), only considering for analysis the patients who did not experience disease recurrence and/or GVHD before day+90 post allo-HSCT.

## 3. Results

### 3.1. IL-2-Activated NK Cell Infusion in Mice

We used preclinical mouse models to assess the safety and efficacy of IL-2-activated NK cell infusion. The NK cells from donor C57BL/6 (CD45.1^+^) mice were isolated, enriched, and activated with IL-2. The distribution, phenotype, and functional activity of CD45.1^+^ IL-2 NK cells after IV injection into congenic C57BL/6 (CD45.2^+^) recipient mice were monitored for 3 days post transfer (Figure 2A). IL-2 activation induced the expression of the CD69, NKG2D, and TRAIL (tumor necrosis factor-related apoptosis-inducing ligand) activation markers on NK cells and increased their responsiveness to YAC-1 tumor cells (Figure 2B and data not shown). Donor-derived (CD45.1^+^) NK cells were detected in the liver, spleen, lung, blood, and lymph nodes of recipient mice for all 3 days post infusion. The relative proportion of donor NK cells in the total NK cell population of each organ was higher in the liver, spleen, and lungs than in the blood and lymph nodes of recipient mice suggesting that these tissues are more prone to promoting NK cell recruitment and/or survival (Figure 2C). However, the frequency of donor NK cells decreased from day 1 to day 3 post transfer in every tissue indicating that activated donor NK cells persist but are relatively short-lived in recipient mice.

Then, we analyzed the phenotype and function of donor IL-2 NK cells over time. The activated phenotype of donor IL-2 NK cells, monitored by measuring CD69 expression, was maintained for at least 3 days in vivo in recipient mice (Figure 2D). The function of donor (CD45.1^+^) and recipient (CD45.2^+^) NK cells was assessed ex vivo 24 h after donor NK cell infusion via stimulation with the tumor cell line YAC-1. Consistent with their activated phenotype, donor-derived NK cells were more responsive than endogenous NK cells. In particular, the frequency of CD107a expression, a surrogate marker of NK cell degranulation, and the frequency of IFN-γ-producing cells were both higher in donor-derived NK cells (Figure 2E). These data indicate that donor IL-2 NK cells survive for several days in recipient mice and maintain their high level of reactivity in vivo after transfer. We did not detect any adverse effect of donor IL-2 NK infusion in recipient mice indicating that this treatment is safe and well tolerated.

### 3.2. Antitumoral Effect of IL-2-Activated NK Cell Infusion in Preclinical Mouse Models

Next, we tested the antitumor effect of IL-2 NK infusion in vivo. We used NKp46-DTR mice, in which NK cells can be depleted via diphtheria toxin (DT) injection, to both discriminate between the roles of recipient and donor NK cells and mimic a clinical situation wherein recipient NK cells are poorly responsive [27]. The NKp46-DTR mice received the following injections: DT, followed the next day by the NK cell-sensitive tumor cell line RMA-S (Figure 3A), then additional DT on days 6 and 15 post-tumor injection. These NK cell-deficient mice were unable to control tumor growth and had a median survival rate of 21.5 days, while mice treated with donor IL-2 NK cells were more resistant to tumor growth and survived longer with half of the mice still alive 70 days after tumor injection (Figure 3B). These results indicate that donor IL-2 NK cell infusions are well tolerated and can have major anti-tumor effects in vivo in a mouse preclinical model.

In this initial model, host NK cells were absent from recipient mice, which does not accurately reflect the clinical situation wherein cancer patients treated by allo-HSCT have hyporesponsive NK cells present [22,23]. Therefore, we created another mouse model with host NK cells present but inhibited in their effector function by HLA-Cw3 ligand present on tumor cells (Figure 3C) and then injected IL-2 NK cells lacking inhibitory receptors for HLA-Cw3. We have previously generated transgenic mice expressing the human inhibitory receptor KIR2DL3 (TgKIR mice) [28,29,30], which were crossed with RAG-1 ^KO^ animals that lack T and B cells (TgKIR-RAG1^KO^) to generate recipients for human tumor cells expressing the MHC class I molecule HLA-Cw3, a ligand for KIR2DL3. The human cell line 221 expressing HLA-Cw3 (221/HLA-Cw3) was then injected into the TgKIR-RAG1 ^KO^ mice. When these mice were treated with PBS, their survival was drastically affected, and they all succumbed in less than 30 days. By contrast, when mice were treated with donor IL-2 NK cells isolated from RAG-1 ^KO^ mice that lack the expression of the KIR2DL3 inhibitory receptor, half of them were still alive 70 days after tumor injection (Figure 3D). These data confirm that infusing fully responsive donor IL-2 NK cells can have very potent antitumor activity in vivo on hematological malignancies.

Altogether, these data provide a proof of concept of the relevance of donor-derived IL-2 NK cell infusion and support the initiation of a phase I clinical trial to assess the safety of donor IL-2 NK cells in humans.

### 3.3. NK Cell Product Manufacturing

After immunomagnetic CD3 depletion and CD56 enrichment, median NK cell recovery was 59% (range: 40–85%) with a median NK cell purity of 95% (range: 80–99%). After 7 days of IL-2 activation, we obtained a median NK cell dose of 4.8 × 10^6^/kg (range: 1.2–11.3), with an NK cell purity of 95% (range: 75–99%) and a median CD3+ cell dose of 0.4 × 10^4^/kg (range: 0–1.5). The NK cell yield after culturing was 52% (range: 5–142%) (Table 1 and Appendix A). We did not observe any T cell expansion after IL-2 activation. Before infusion into patients, we verified that all cellular products fulfilled the following release criteria: absence of bacterial contamination, total nucleated cell viability >70%, and CD3+ T cell counts <5 × 10^4^/kg.

### 3.4. Characterization of Donor IL-2-Activated NK Cells

Donor IL-2-activated NK cells were analyzed and qualified before infusion into the patients. Compared to resting NK cells from the same donors, IL-2 NK cells expressed higher levels of the activating receptors NKp30, NKp46, NKG2D, DNAM-1, and CD16 (Figure 4A) [31,32,33]. In addition, IL-2 NK cells also expressed the inducible activating receptor NKp44 as well as the activation marker CD69 (Figure 4B) [34,35], while TRAIL, which is not expressed on resting NK cells, was also strongly up-regulated on NK cells after 7 days of IL-2 activation (Figure 4B). By contrast, the expression of the MHC class I-specific receptors KIR2DL1/S1 and KIR2DL2/S2/L3 was significantly decreased, whereas CD94 and NKG2A expression was increased (Figure 4C).

We then performed a multi-parameter flow cytometry analysis to evaluate the frequency of donor NK cell expression of the different KIRs and/or CD94/NKG2A receptors. We found that the frequency of NK cells expressing none of the inhibitory receptors tested was decreased whereas the frequency of cells expressing NKG2A alone, without one of the other tested KIRs, was increased from 31.5% (+/− 3.2%) to 47.4% (+/− 4.5%) (Figure 4D). These data demonstrate that donor IL-2 NK cells harbor an activated phenotype and express high levels of activating receptors. However, the frequency of cells expressing the inhibitory receptor CD94/NKG2A was also increased.

We then analyzed the effector functions of donor IL-2 NK cells compared to the corresponding resting NK cells from the same donors. Various stimulatory conditions were used to assess NK cell responsiveness including stimulation with the tumor cell line K562, with anti-CD16 monoclonal antibodies, with a mix of IL-12 and IL-18 inflammatory cytokines, or with PMA and ionomycin (Figure 4E). In all of these conditions, NK cell IFN-γ production as well as NK cell degranulation (CD107a and CD107b expression), a surrogate marker of NK cell cytotoxicity, were higher in IL-2 NK cells compared to resting NK cells (Figure 4E).

These data suggest that IL-2 activation of donor NK cells increases their responsiveness to multiple stimuli, increasing the potential for more potent antitumor activities than resting NK cells. We therefore decided to use these donor IL-2 NK cell preparations to treat the patients included in the clinical trial.

### 3.5. Patients Included in the Clinical Trial

With a median age of 59 years (range: 38–68), 16 patients received donor-derived IL-2 NK cell infusions within a median time of 89 days (range: 61–119) after allo-HSCT (Figure 1). The phase I clinical trial included the following hematologic diseases: acute myeloid leukemia (*n* = 6), lymphoma (*n* = 5), myelodysplastic syndrome (*n* = 1), acute lymphoblastic leukemia (*n* = 1), primary myelofibrosis (*n* = 1), and myeloma (*n* = 2). The DRI was intermediate for all patients and eight patients (50%) had hematopoietic cell transplantation comorbidity index scores ≥3. All patients were in morphologic complete remission at the time of NK cell infusion. Among AML patients, three patients did not have identified molecular markers for minimal residual disease (MRD) monitoring, one had *NPM1* and *FLT3-ITD* mutations (patient number 11), one had *RUNX1-RUNX1T1* gene fusion (patient number 14), and one had *BCR-ABL1* gene fusion (patient number 12, Table 2). Only the one with *NPM1* mutation was in molecular remission at the time of IL-2 NK cell infusion. After NK cell infusion, median follow-up was 37 months (range: 29–66).

As no DLT was observed, the first phase of dose escalation included nine patients (three per dose level) and the extension phase included seven additional patients at the highest dose level (level 3). All three patients at dose level 1 (1.0 × 10^6^/kg) received the targeted IL-2 NK cell dose. At dose level 2 (5.0 × 10^6^/kg), one patient was given the expected IL-2 NK cell dose while the two remaining patients received 4.0 and 4.4 × 10^6^/kg. Among the 10 patients included at dose level 3 (5.0 to 50 × 10^6^/kg), the median infused IL-2 NK cell dose was 4.8 × 10^6^/kg (range: 1.3–10); five patients were unable to receive the minimal target IL-2 NK cell dose of 5.0 × 10^6^/kg. Characteristics and outcomes of all patients are detailed in Table 2.

During the 30-day DLT assessment period, one patient developed idiopathic facial paralysis with spontaneous resolution. No additional early adverse event was observed. After the assessment period, four patients developed chronic GVHD at day+31 (severe), day+72 (moderate), day+192 (mild), and day+346 (moderate) post IL-2 NK cell infusion, corresponding to days 142, 155, 267, and 446 after allo-HSCT, respectively (Table 2). The three patients with moderate or severe chronic GVHD received systemic steroid-based treatment. At last follow-up, all four patients had complete remission (CR) of chronic GVHD and were not given any immunosuppressive treatment. We observed one non-relapse death due to idiopathic pneumonia 2 months after IL-2 NK cell infusion (UPN10, dose 2.7 × 10^6^/kg). Among the four patients who relapsed (3 AML and 1 CLL), two died from their disease and two were in complete response after salvage therapy. At last follow-up, 11 out of 16 treated patients were alive in CR without GVHD features or immunosuppressive treatment.

We compared the outcome of the patients included in the DLI-NK trial to a historical control cohort of patients treated in the same center who did not receive IL-2 NK (*n* = 61), including 27 AML, 16 lymphoma, 12 myeloma, 5 ALL, and 1 CLL. All of them were considered at intermediate risk according to the DRI, received similar allo-HSCT procedures than the IL-2 NK group (MSD, reduced intensity conditioning regimens), and were disease and GVHD free at day+90 after allo-HSCT. Compared with this control cohort, no increase in 2-year GVHD was observed after IL-2 NK cell infusion (control vs. IL-2 NK: 36% vs. 25%, *p* = 0.373, Figure 5A). There was no significant difference in 2-year PFS (control vs. IL-2 NK: 60% vs. 75%, *p* = 0.341, Figure 5B) and OS (control vs. IL-2 NK: 73% vs. 88%, *p* = 0.420).

### 3.6. NK Cell Immune Monitoring After Infusion

Patients with available samples for all time points were included in these analyses. NK B, T, and γδ-T cell frequencies in blood samples did not significantly change between day+0 and day+1 after IL-2 NK cell infusion (Figure 6A). We did not observe significant changes in NK cell phenotype or function between day+0 and day+1 (Appendix A). We observed a trend towards improved NK cell degranulation (CD107a, *p* = 0.020) and cytokine production (IFN-γ, *p* = 0.027; TNF-α, *p* = 0.008; MIP-1β, *p* = 0.129) at day+9 compared to day+0 when co-cultured in vitro with K562 cell lines (Figure 6B). We then performed longitudinal phenotypic analyses of the nine patients who completed the full sequence of blood sample collections from day+0 to day+180. As previously described after allo-HSCT [22,23,26], we observed that NK maturation recovered progressively over time alongside the decrease in NKG2A expression (*p* = 0.039) and concomitantly to an increase in CD57 expression (*p* = 0.016) (Figure 6C). No differences in the expression of KIRs and NK cell-activating receptors were observed after NK cell infusion (data not shown). In addition, we performed longitudinal analyses of NK functions (degranulation and cytokine production) on blood samples collected on day+0, day+1, day+9, day+16, and day+30 after IL-2 NK cell infusion. Among the seven patients for whom we had sufficient cells to perform said analysis, we observed a trend towards progressive recovery of NK functions after NK cell infusion, as expected after allo-HSCT (Figure 6D). Thus, we were unable to detect any specific phenotypic or functional NK cell dysfunction that could be related to the infusion of highly activated donor-derived NK cells.

## 4. Discussion

Our preclinical data in the mouse model showed that NK cells activated ex vivo with high doses of IL-2 harbor a very active profile, both in terms of phenotype and function. They also demonstrated that IL-2 NK cell infusions in recipient mice can be not only safe but also efficient against tumor cells. Although we have been able to detect donor IL-2 NK cells 3 days after infusion, it is not clear how long they can persist in this mouse model. This underlines the need to better investigate the outcome of infused NK cells and the potential benefit of using multiple NK infusions to increase the efficacy of the treatment.

Our phase I clinical trial demonstrates the feasibility and safety of donor-derived ex vivo IL-2-activated NK cell infusion after MSD allo-HSCT. No DLT was observed after dose escalation nor were any severe long-term adverse events related to IL-2 NK cell infusion. Notably, only four patients (25%) developed GVHD (all were chronic forms, three required treatment), all of which were in remission and did not receive any immunosuppressive treatment. Of the four GVHD cases, two occurred more than 6 months (day +192 and +346) after NK cell infusion and were therefore most likely unrelated. In the other two patients, GVHD occurred sooner after IL-2 NK cell infusion, on day+31 and day+72. In these cases, causality between IL-2 NK cell infusion and GVHD cannot be excluded. However, it is important to note that these patients received IL-2 NK cell infusions on day+111 and day+83 after transplantation, meaning that GVHD actually occurred within the first 6 months following allo-HSCT (day+142 and day+155). Chronic GVHD is frequently reported in the post-transplantation period during the follow-up of patients receiving MSD transplantation after an ATG-based reduced-intensity conditioning regimen, even in the absence of any immune intervention (cumulative chronic GVHD incidence: 15% to 30% [36,37]). Our comparison with a historical cohort from the same center suggests that IL-2 NK cell infusion does not increase the incidence of GVHD after MSD allo-HSCT. In addition, we previously observed chronic GVHD incidence rates of 37% to 51% after conventional prophylactic DLI in different diseases and donor types [8,9]. Thus, our data indicate that donor IL-2 NK cell infusions do not induce GVHD at a higher rate than conventional DLI. Moreover, we observed promising PFS and OS (75% and 88% at 2 years, respectively), which suggest potential efficacy. However, these comparisons have their limits as they suffer from methodological bias due to the low number of patients included in the study and their numerous disease types. The results obtained in our IL-2 NK cohort are comparable to the findings of other conventional prophylactic DLI studies [8,9,38,39,40].

Immune monitoring did not reveal major significant changes in patient NK cell phenotype and function after IL-2 NK cell infusion, suggesting that the infusion of donor-derived highly activated NK cells did not alter the kinetics of global NK cell recovery after allo-HSCT. As expected after allo-HSCT, we observed progressive NK cell maturation and function recovery over time (Figure 6), but it is unclear to what extent NK infusion participated in this evolution. These analyses were limited in part by the low number of patients with enough cells to analyze the entire sequence over time. Furthermore, while taking into account the low number of infused NK cells and the lack of any means to track infused NK cells in vivo, the monitoring of NK cell post-infusion data mostly characterizes donor-derived pre-existing NK cells. Yet, our preclinical mouse model data suggest that IL-2-activated NK cells can migrate into tissues, persist for at least 3 days, and retain their hyperactivity in vivo after infusion. Finally, our clinical data suggest that donor-derived IL-2 NK cell infusions are well tolerated after allo-HSCT with little GVHD, no severe disruption of NK cell recovery, and promising survival.

Beyond clinical results, our study confirms the feasibility of NK cell manufacturing to produce pure and highly active NK cells. However, the NK cell yield was relatively low at the end of the 7-day activation process, resulting in a failure to infuse sufficient numbers of NK cells in dose level 3; 5 out of the 10 patients at this dose level actually received NK cell doses of ≥5 × 10^6^/kg. It is possible that a shorter culture time results in higher yield, as it was reported by Miller’s group (overnight IL-2 incubation, median NK cell dose of 8.5 × 10^6^/kg) [41]. Moreover, using ex vivo expanded NK cells rather than IL-2 activated NK cells could eventually allow for the production of sufficient NK cells for multiple infusions, especially when allogeneic feeder cells are used to trigger NK cell activation and proliferation according to the extensive review by Granzin et al. [42]. Various protocols have been developed to produce high amounts of activated allogeneic NK cells, with the expansion ratio reaching 100- to 1,000-fold increases [42,43,44,45]. However, the degree of activation after feeder-based expansion remains unclear compared to the high activation phenotype of IL-2 NK cells. In addition, protocols favoring the expansion of memory-like NK cells may be appealing to enhance activation and in vivo persistence after infusion [46].

To date, only Ciurea and colleagues have reported the clinical use and safety of ex vivo expanded donor-derived NK cells after allo-HSCT in a phase I trial (K562 feeder cells transduced with membrane-bound IL-21 and 4-1BBL) [47]. Similar approaches were reported in expanding cord-blood-derived NK cells to produce anti-CD19 allogeneic chimeric antigen receptor NK cells, highlighting the wide use of NK cell expansion processes [48].

The NK cells obtained after 7 days of IL-2 activation presented highly activated profile compared to resting NK cells, with a significantly higher expression of activating NK cell receptors resulting in increased degranulation (CD107a/b) and IFN-γ production in functional assays (Figure 4). However, we also observed a significant increase in NKG2A^+^ KIR^−^ NK cells while the proportions of NK cells harboring other inhibitory receptor combinations were similar before and after IL-2 activation (Figure 4D). This increased NKG2A expression may limit GVL effect against HLA-E^+^ tumors. It was previously shown that anti-NKG2A blocking monoclonal antibodies (monalizumab) can overcome this limitation in different settings, notably when ex vivo IL-2 activated NK cells are co-cultured with HLA-E^+^ targets [24,49,50]. Taken together, these observations provide a strong rationale for the use of anti-NKG2A-blocking monoclonal antibodies to improve the efficacy of donor-derived NK cells against HLA-E-positive cancers. In addition, the relevance of combining NK cell infusion and inhibitory receptor blockade is reinforced by our preclinical mouse model data, which demonstrated the potent antitumor role of IL-2 NK infusion when inhibitory receptors are not engaged by tumor targets. Therefore, we are currently conducting a phase I trial of monalizumab (NCT02921685) as maintenance therapy after allo-HSCT that should provide preliminary data in humans before future donor-derived NK cell infusion combinations.

## 5. Conclusions

In conclusion, our phase I trial demonstrates the clinical and biological safety of donor-derived IL-2 NK cell infusion as prophylactic cell-based immunotherapy after MSD allo-HSCT. In accordance with clinical and biological rationales [11,14,51,52], the low incidence of GVHD compared to conventional DLI suggests that IL-2-activated NK cell infusion may reduce the incidence of GVHD. In addition, survival was promising, supporting the design of future clinical trials evaluating the efficacy of these treatments.

## Figures and Tables

**Figure 1 cancers-13-02673-f001:**
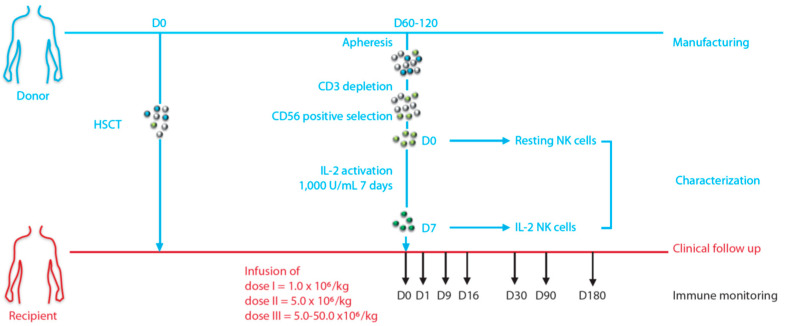
Design of the clinical trial: DLI-NK NCT01853358.

**Figure 2 cancers-13-02673-f002:**
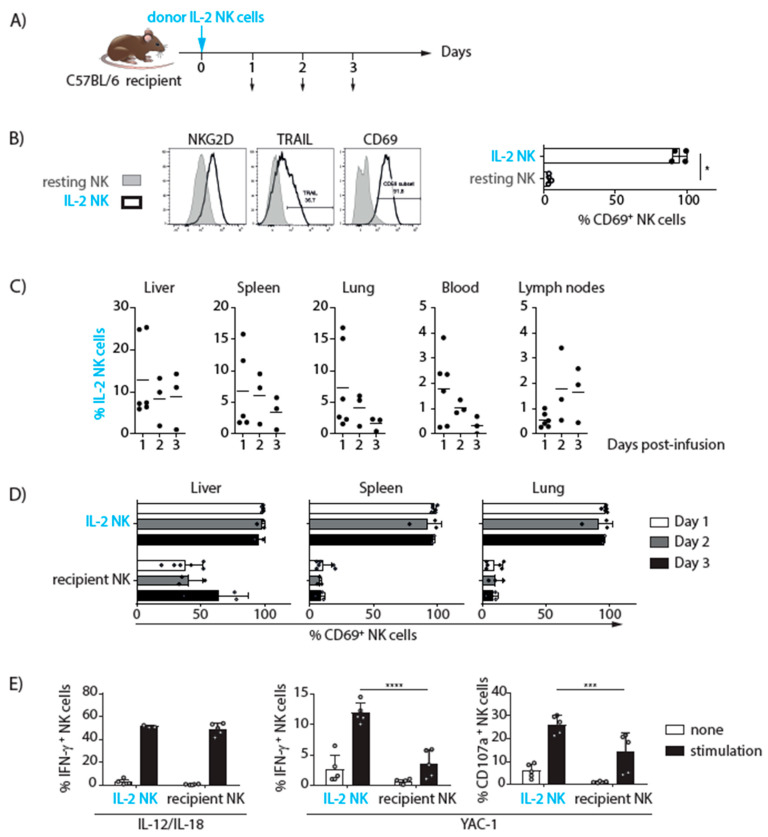
Monitoring of donor IL-2 NK cell phenotype and function in a mouse model. (**A**) Experimental scheme: donor IL-2 NK cells expressing the marker CD45.1^+^ were injected into congenic CD45.2^+^ recipient mice and their distribution, phenotype, and functional activity were monitored on day 0 before infusion and days 1, 2, and 3 post-infusion by flow cytometry. (**B**) Representative flow cytometry histograms showing the expression of NKG2D, TRAIL, and CD69 in resting and IL-2 NK cells before infusion (left panel). Percentages of CD69 expression in resting and IL-2 NK cells before infusion (right panel). Statistical analysis was performed with a Mann–Whitney test. ** p < 0.05*. (**C**) Percentages of donor IL-2 NK cells out of total NK cell populations in liver, spleen, lung, blood, and lymph nodes at the indicated post-infusion time points. (**D**) Percentages of CD69^+^ cells in donor (CD45.1^+^) and recipient (CD45.2^+^) NK cells in liver, spleen, and lung at days 1, 2, and 3 post-infusion. (C-D) Data were pooled from 3 experiments including one experiment with 3 mice analyzed at day 1, one experiment with 1 mouse analyzed at each time point and one experiment with 2 mice analyzed at each time point. Statistical analysis was performed with a one-way (**C**) or two-way ANOVA test (**D**). (**E**) Splenocytes from recipient mice were harvested 24 h post-infusion and activated ex vivo with or without a mix of IL-12 and IL-18 or the YAC-1 tumor cell line. The percentages of IFN-γ^+^ and CD107a^+^ donor (CD45.1^+^) and recipient (CD45.2^+^) NK cells are shown. Data were pooled from 2 experiments including one experiment with 1 mouse analyzed at each time point and one experiment with 2 mice analyzed at each time point. Statistical analysis was performed with a two-way ANOVA and Sidak’s multiple comparaison test. *** *p* = 0.0004 and **** *p* < 0.0001.

**Figure 3 cancers-13-02673-f003:**
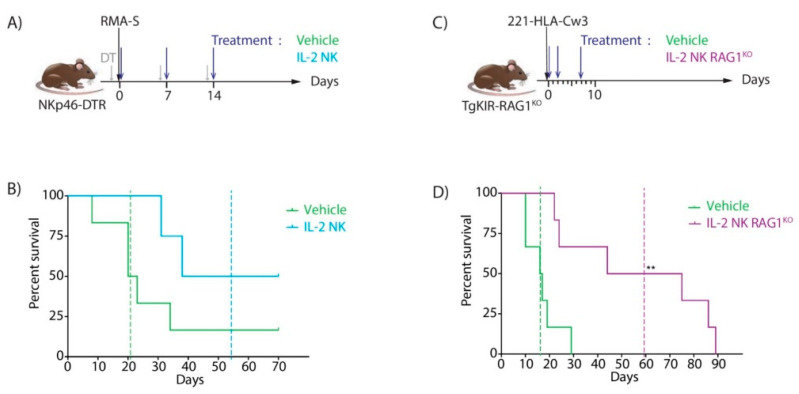
Anti-tumor effect of donor IL-2 NK cells in mouse models. (**A**) Experimental scheme: NKp46-DTR mice were treated by diphtheria toxin (DT) 1 day before and on days 6 and 13 after injection of the NK cell-sensitive tumor cell line RMA-S (day 0). Donor IL-2 NK cells or vehicle only were infused on days 0, 7, and 14. (**B**) Survival rates after RMA-S injection in NK cell-deficient mice treated with IL-2 NK cells (blue line) or vehicle only (green line). Median survival of IL-2 NK-treated (dotted blue line) and control group (dotted green line). (**C**) Experimental scheme: 221/HLA-Cw3 human tumor cells were injected in TgKIR-RAG1^KO^ mice on day 0 and treated with either vehicle only or donor IL-2 NK cells from RAG1^KO^ mice on days 0, 2, and 7. (**D**) Survival rate of tumor-bearing TgKIR-RAG1^KO^ mice upon treatment with either PBS (green line) or IL-2 NK cells (purple line). Median survival for each group is represented by dotted lines. Log rank test, *** p =* 0.0042.

**Figure 4 cancers-13-02673-f004:**
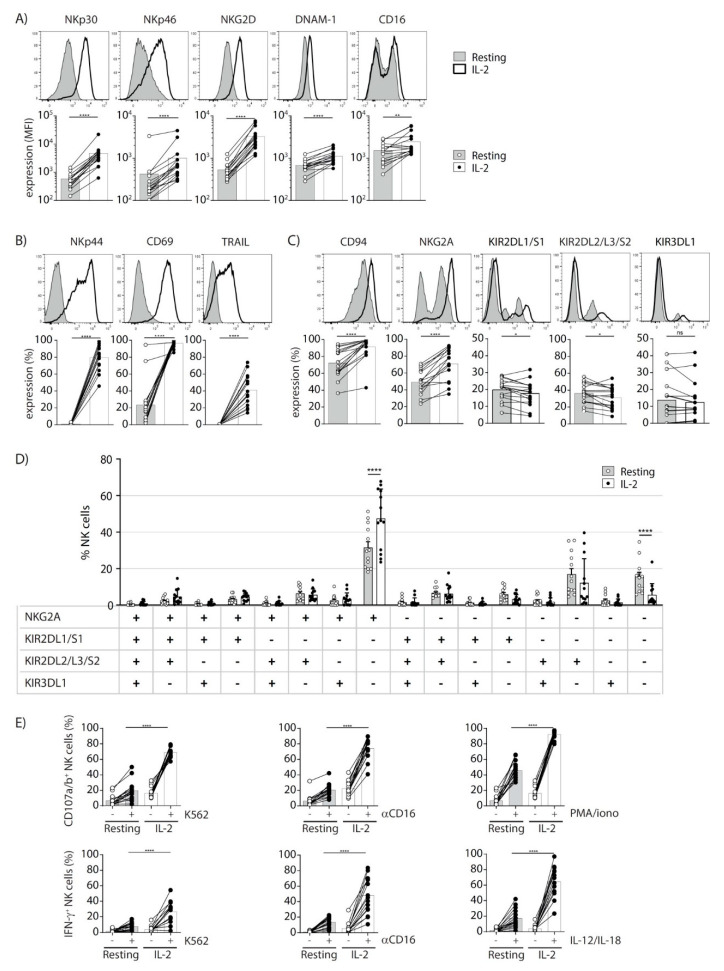
Phenotypic and functional characterization of resting and IL-2-activated donor NK cells. (**A**–**C**) Representative flow cytometry histograms (upper panels) and expression levels for each donor (mean fluorescence intensity, MFI, or % as indicated, lower panels), illustrating (**A**) NKp30, NKp46, NKG2D, DNAM-1, and CD16; (**B**) NKp44, CD69, and TRAIL; (**C**) CD94, NKG2A, KIR2DL1/S1, KIR2DL2/L3/S2, and KIR3DL1 expression before (resting) or after IL-2 activation. (**D**) The percentages of NK cells expressing the indicated markers for resting (grey histograms) and IL-2 activated NK cells (white histograms). (**E**) Functional analysis of resting versus IL-2-activated NK cells from the same donors. NK cells were stimulated or not during 4 h with the tumor cell line K562 (left panels), anti-CD16 antibodies (middle panels), and PMA and ionomycin or IL-12 and IL-18 (right panels). CD107a/b expression (upper panels) and IFN-γ production (lower panels) are shown for each donor. Statistical analysis was performed with a Wilcoxon test, **p* = 0.0110, ***p* = 0.0013, **** *p* < 0.0001; ns, no significant (**A**–**C**), a two-way ANOVA test and Sidak’s multiple comparaison test, **** *p* < 0.0001, only significant *p* values are indicated (**D**) and a one-way ANOVA test, only the *p* values comparing the responsiveness of resting versus IL-2 activated NK cells are indicated, **** *p* < 0.0001 (**E**).

**Figure 5 cancers-13-02673-f005:**
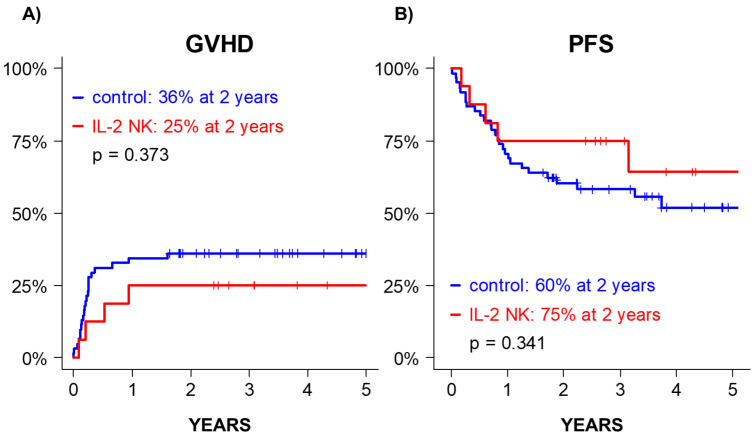
Outcome after IL-2 NK cell infusion. GVHD (**A**) and PFS (**B**) of the cohort of patients treated with donor IL-2 NK cells (patients in the clinical trial, blue line) vs. historical control cohort (red line).

**Figure 6 cancers-13-02673-f006:**
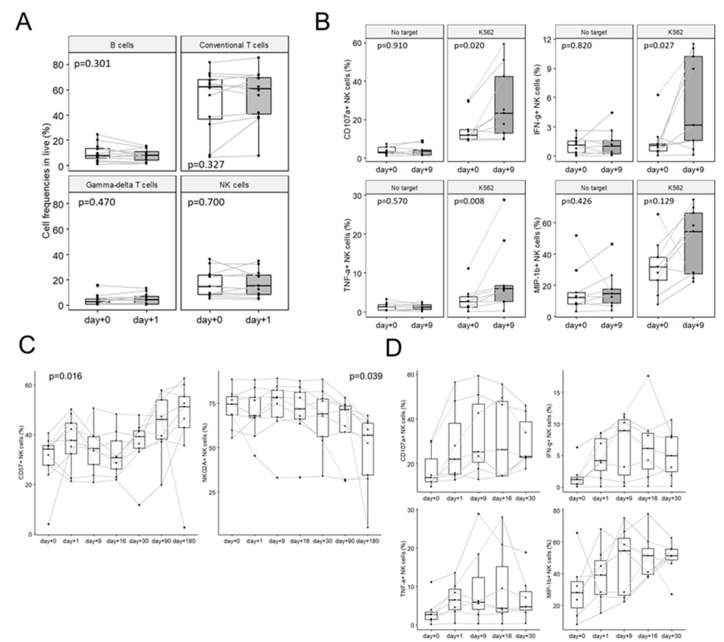
Immune monitoring by FACS after IL-2 NK infusion. (**A**) B, T, γδ-T, and NK cell frequencies between day+0 and day+1 after IL-2 NK cell infusion (*n* = 12). (**B**) NK cell functional assays before (day+0) and after (day+9) IL-2 NK infusion: CD107a, IFN-γ, TNF-α, and MIP-1α expression on NK cells after culturing PBMCs with or w/o K562 at an E:T ratio of 10:1 for 4 h (*n* = 9). (**C**) Longitudinal immune monitoring of the NK cell maturation markers NKG2A and CD57 (*n* = 9). (**D**) Longitudinal analyses of CD107a, IFN-γ, TNF-α, and MIP-1β expression on NK cells. PBMCs were co-cultured for 4 h with or without (not shown) K562 at an E:T of 10:1 (*n* = 7). Statistical tests were paired Wilcoxon (A and B) and paired Kruskal–Wallis (C) tests.

**Table 1 cancers-13-02673-t001:** Cell product composition in NK cells at apheresis and before and after 7-day IL-2 activation.

	NK Cells (10^6^/kg)		NK Cell Purity (%)		NK Cell Yield (%)	
	Median	(min–max)	Median	(min–max)	Median	(min–max)
Before isolation	18.8	(4.9–53.4)	8%	(3–13)	-	-
After isolation	11.1	(4.1–31.3)	95%	(80–99)	59%	(40–85)
After culture in IL-2	4.8	(1.2–11.3)	95%	(75–99)	52%	(5–142)

**Table 2 cancers-13-02673-t002:** Patient characteristics and outcomes.

UNP	Gender	Age	Diagnosis	DRI	IL-2 NK Infusion after Allo-HSCT	Dose Level (10^6^/kg)	Dose Received (10^6^/kg)	GVHD	Other Adverse Events	Outcome
#01	F	47	AML	Int	day+61	I = 1.0	1.0		Facial paralysis (d+15); resolved	Relapse (37.8m) Death (41.8m)
#02	F	57	T-NHL	Int	day+78	I = 1.0	1.0			Alive in CR (66.3m)
#03	M	66	PMF	Int	day+82	I = 1.0	1.0	Moderate cGVHD (d+346); resolved		Alive in CR (51.4m)
#04	M	43	HL	Int	day+75	II = 5.0	4.0	Mild cGVHD (d+192); resolved	Encephalitis (d+118); resolved	Alive in CR (64.5m)
#05	M	47	T-NHL	Int	day+85	II = 5.0	4.4		Auto immune thyroiditis (d+87); resolved	Alive in CR (52.1m)
#06	F	55	T-ALL	Int	day+86	II = 5.0	5.0			Alive in CR (63.1m)
#07	M	48	AML	Int	day+96	III = 5.0–50.0	4.0			Alive in CR (28.7m)
#08	M	38	AML	Int	day+78	III = 5.0–50.0	4.6			Relapse (3.9m) Death (4.2m)
#09	M	65	Myeloma	Int	day+97	III = 5.0–50.0	8.5			Alive in CR (45.7m)
#10	M	68	Myeloma	Int	day+103	III = 5.0–50.0	2.7		Idiopathic pneumonia (d+34); fatal	Death (2.1m)
#11	F	60	AML	Int	day+106	III = 5.0–50.0	4.5			Alive in CR (36.9m)
#12	F	52	AML	Int	day+98	III = 5.0–50.0	10.0			Relapse (7.3m) Alive (37.1m)
#13	F	68	MDS	Int	day+111	III = 5.0–50.0	1.3	Severe cGVHD (d+31); resolved		Alive in CR (30.7m)
#14	F	67	AML	Int	day+119	III = 5.0–50.0	5.0			Alive in CR (31.8m)
#15	F	63	LL	Int	day+83	III = 5.0–50.0	6.0	Moderate cGVHD (d+72); resolved		Alive in CR (33.0m)
#16	M	61	ProLL	Int	day+92	III = 5.0–50.0	7.3			Relapse (9.9m) Alive (39.6m)

ALL: acute lymphoblastic leukemia; AML: acute myeloid leukemia; CR: complete remission; DRI: disease risk index; HL: hodgkin lymphoma; LL: lymphoblastic lymphoma; MDS: myelodysplastic syndrome; NHL: non hodgkin lymphoma; PMF: primary myelofibrosis; ProLL: prolymphocytic leukemia.

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
