# Peer review of "Phase I Trial of Prophylactic Donor-Derived IL-2-Activated NK Cell Infusion after Allogeneic Hematopoietic Stem Cell Transplantation from a Matched Sibling Donor"

_cancers, 2021, doi:10.3390/cancers13112673_

Round 1

Reviewer 1 Report

Overall a very well done study with both pre-clinical mouse model and clinical trial. However I do have following questions/ suggestions for the authors to help further improve this manuscript. 

  1. What was the rationale for doing mouse work as there have been several previous clinical trials of NK cells not sure they needed to do any pre clinical mouse work, please address this (add justification for doing pre-clinical work) in the discussion section.
  2. The NK cells persisted only few days in their congeneic mouse model, why? Could it be from cytokine addiction? It would be nice to compare in vivo survial of these IL-2 activated Nk cells with cytokine induced memory like (CIML) NK cells in view of several recent publications  (romee et al, Blood 2012, Ni et al JEM 2013 and Romee et al, Science TM 2016). Also it might be worth comparing in vivo persistence / expansion with and without IL-2 activation (to address cytokine addiction issue).
  3. In the second mouse model, how long did the adoptively transferred NK cells persist?
  4. In the figure 4 they show increased expression of NKG2A, would this limit in vivo GvL effect of these cells, if so what would be their future strategy to mitigate this issue, kindly add it to the discussion section.
  5. For the trial, why did they include non myeloid malignancy patients as most of the previous studies have shown NK cell mediated GvL effect only in myeloid malignancies?
  6. Do they have any information on the underlying mutational profiles and MRD of these patients?
  7. For PFS/ OS suggest comparing to a contemporaneous cohort from the same center, could also do propensity score matching
  8. What was the absolute lymphocyte count in these patents right before NK cell infusion? Did the ALC affect expansion and persistence of the adoptively transferred NK cells. This could be the reason for not seeing much NK cell expansion. Need to address this issue in their discussion. Also for future efforts would they consider adding some lymphodepletion? Furthermore, what about IL-2 or low dose IL-15 after adoptive transfer? Please address these critical issues in the discussion section.
  9. Was their immunophenotype any different from contemporaneous non NK cell infusion cohort patients at the same post HCT time points?

Reviewer 2 Report

In this manuscript, the authors presented a NK cell based immunotherapy as a potential maintenance therapy to overcome relapses after allogeneic hematopoietic stem cell transplantation in patients with hematologic malignancies.  Results presented indicate that donor-derived IL-2 NK cells are safe (low incidence of GVHD compared to conventional DLI) and potentially efficacious (high PFS and OS) when used as prophylactic therapy after allo-HSCT. In addition to the promising early clinical data, observations from the preclinical models also opened the door for the possibility of combination therapy with inhibitory receptor blockage to further boost antitumor efficacy.  Overall, the topic is highly relevant and should be of interest to the readers. Studies presented in the manuscript were also well thought out and appropriately controlled. Given this, I would recommend this manuscript to be accepted after minor revision.

  1. Line 73, there was some formatting issue with IFN-gamma, it's not showing up correctly on my end.
  2. Figure 6C, y axis was cut off.
  3. Line 458, how is the PFS and OS observed compared to allo-HSCT alone or conventional DLI? 

Round 2

Reviewer 1 Report

The authors have addressed all my comments and questions.